# Molecular and Antioxidant Characterization of *Opuntia robusta* Fruit Extract and Its Protective Effect against Diclofenac-Induced Acute Liver Injury in an In Vivo Rat Model

**DOI:** 10.3390/antiox12010113

**Published:** 2023-01-03

**Authors:** Gloria Stephanie Villa-Jaimes, Han Moshage, Francisco Javier Avelar-González, Herson Antonio González-Ponce, Manon Buist-Homan, Fidel Guevara-Lara, Esperanza Sánchez-Alemán, Sandra Luz Martínez-Hernández, Javier Ventura-Juárez, Martín Humberto Muñoz-Ortega, Ma. Consolación Martínez-Saldaña

**Affiliations:** 1Department de Morfología, Centro de Ciencias Básicas, Universidad Autónoma de Aguascalientes, Aguascalientes 20100, Mexico; 2Department of Gastroenterology and Hepatology, University Medical Center of Groningen, University of Groningen, 9713 ZP Groningen, The Netherlands; 3Department of Laboratory Medicine, University Medical Center Groningen, University of Groningen, 9713 ZP Groningen, The Netherlands; 4Departamento de Fisiología, Centro de Ciencias Básicas, Universidad Autónoma de Aguascalientes, Aguascalientes 20100, Mexico; 5Instituto Tecnológico de Aguascalientes, Aguascalientes 20256, Mexico; 6Departamento de Química, Centro de Ciencias Básicas, Universidad Autónoma de Aguascalientes, Aguascalientes 20100, Mexico; 7Unidad de Medicina Familiar 8, Instituto Mexicano del Seguro Social (IMSS), Aguascalientes 20180, Mexico; 8Departamento de Microbiología, Centro de Ciencias Básicas, Universidad Autónoma de Aguascalientes, Aguascalientes 20100, Mexico

**Keywords:** oxidative stress biomarkers, bioactive compounds, antioxidant response, dietary antioxidants, diclofenac, liver failure

## Abstract

A molecular characterization of the main phytochemicals and antioxidant activity of *Opuntia robusta* (OR) fruit extract was carried out, as well as an evaluation of its hepatoprotective effect against diclofenac (DF)-induced acute liver injury was evaluated. Phenols, flavonoids and betalains were quantified, and antioxidant characterization was performed by means of the ABTS^•+^, DPPH and FRAP assays. UPLC-QTOF-MS/MS was used to identify the main biocompounds present in OR fruit extract was carried out via. In the in vivo model, groups of rats were treated prophylactically with the OR fruit extract, betanin and *N*-acteylcysteine followed by a single dose of DF. Biochemical markers of oxidative stress (MDA and GSH) and relative gene expression of the inducible antioxidant response (*Nrf2*, *Sod2*, *Hmox1*, *Nqo1* and *Gclc*), cell death (*Casp3*) and DNA repair (*Gadd45a*) were analyzed. Western blot analysis was performed to measure protein levels of Nrf2 and immunohistochemical analysis was used to assess caspase-3 activity in the experimental groups. In our study, the OR fruit extract showed strong antioxidant and cytoprotective capacity due to the presence of bioactive compounds, such as betalain and phenols. We conclude that OR fruit extract or selected components can be used clinically to support patients with acute liver injury.

## 1. Introduction

Historically, natural products of plant origin have been used worldwide for medicinal purposes. In Mexico, since pre-Hispanic times, foods, such as corn, maguey and nopal, are part of the population’s diet [1,2]. Interest in studying food consumption and its beneficial effects on health has recently increased [3].

The genus *Opuntia* spp. is distributed naturally in the southern United States and Latin America and was introduced in Asia, South Africa, Ethiopia, Australia and countries of the Mediterranean basin, such as Italy, Spain and Greece [4]. Mexico is considered the center of the origin and diversity of the genus [5]. The fruits of various species of *Opuntia* spp. contain compounds with biological activity, such as vitamins B1, A, E and C [6,7], flavonoids (isorhamnetin, kaempferol, quercetin) [6], phenolic compounds (ferulic acid and synapoyl-diglycoside) [6] and pigments (betacyanins and betaxanthins) [8,9].

Likewise, experimental in vivo [10], in vitro and clinical studies have shown that phytochemicals from the fruit of *Opuntia* spp. have activities, such as: cholesterol-lowering effects (pectin of OR) [11], antiatherogenic effects (betanin and indicaxanthin from *O. ficus-indica*) [12,13], anticarcinogenic effects (phenolics, flavonoids, betalains from *Opuntia* spp.) [14,15,16], antigenotoxic and hepatoprotective effects (betacyanins, phenolics and flavonoids in *O. ficus-indica*, *O. streptacantha* and *O. robusta*, respectively) [17,18]. The biological effects of *Opuntia* spp. fruit consumption have been associated with the antioxidant capacity of phytochemicals via their interaction with different prooxidant mechanisms and the inducible antioxidant response [18].

The *O. robusta* (OR) species is widely distributed in Mexico, with cultivars in South Africa and the Mediterranean basin [19]. Its fruit (prickly pear) has a purple coloration and contains phenolic compounds, such as p-hydroxy benzoic acid, ferulic acid, pyrogallol [20], vitamins C and E [7] and pigments (betalains, mainly betacyanins) [21].

Diclofenac (DF) is the most widely prescribed over-the-counter (OTC) non-steroidal anti-inflammatory drug (NSAID) in the world [22] and together with ibuprofen accounts for 40% of sales of NSAIDS for osteoarthritis [23] in people aged between 65–85 years with multiple morbidities [24]. It is also prescribed to treat chronic pain related to rheumatoid arthritis, ankylosing spondylitis and acute muscle pain [25]. Its frequent and/or excessive consumption is related to the development of adverse effects in the gastrointestinal tract and liver [26]. Drug-induced liver injury (DILI) is an adverse reaction caused by an increase in cellular oxidative stress, which damages hepatocytes and other liver cells, as a result of drug biotransformation in the liver [27]. This toxicity may be due to the interaction between the drug and special risk factors of the patient (e.g., age, primary pathologies, genetic factors), known as idiosyncrasy (iDILI), or may be dose-dependent, as seen in intrinsic DILI [28,29]. According to Teschke & Danan (2020) [30], diclofenac ranks fifth in the worldwide top ranking of drugs causing DILI cases with a causality assessment performed based on the Roussel Uclaf Causality Assessment Method (RUCAM). Scientometric research reported that of 3312 cases, 46 corresponded to diclofenac iDILI [31,32,33]. Diclofenac acute liver injury is mediated mainly by the isoforms CYP2C9 and CYP3A4 [34]. It generates reactive oxygen species (ROS), reactive nitrogen species (RNS) and electrophilic compounds, such as iminoquinones, which enter redox cycles and produce mitochondrial injury in hepatocytes [35,36,37].

The protective effect of OR fruit extract against the hepatotoxicity induced by thioacetamide (TAA) [7], acetaminophen (APAP) [17,38] and diclofenac [39] has been previously reported, where biomarkers of liver injury were monitored, such as the aspartate aminotransferase (AST) and alanine aminotransferase (ALT) enzymes. Moreover, effects on the expression and/or activation of nuclear factor kappa-light-chain-enhancer of activated B cells (NF-κB) [40], mediated by indicaxanthin, and nuclear factor erythroid 2-related factor 2 (Nrf2), mediated by betanin [41], have also been reported.

Therefore, the aim of this study was to complement the characterization of the biocomponents of OR and its antioxidant properties against DF-induced acute liver injury.

## 2. Materials and Methods

### 2.1. Chemicals

Gallic acid monohydrate #398225, (+)-catechin hydrate #C1251, DPPH (2,2-diphenyl-1-picrylhydrazyl #D9132), Trolox ((±)-6-hydroxy-2,5,7,8-tetramethylchroman-2-carboxylic acid #238813), ABTS^•+^ (2,2′-azino-bis(3-ethylbenzothiazoline-6-sulfonic acid diammonium salt #A1888), diclofenac sodium salt #D6899, N-acetyl-L-cysteine #A7250 and betanin #901266 were provided by Sigma-Aldrich Chemicals (St. Louis, MO, USA).

### 2.2. Plant Material and Sample Preparation

Samples of wild *Opuntia robusta* fruits were collected from randomly selected plants in a semi-arid region east of Aguascalientes, México (21°47′04.7″ N 102°06′40.1″ O). Fruits were washed and disinfected, and the peel was separated from the fruit. Sample preparation was carried out following the method described previously [38]. The fruit juice was centrifuged, filtered, lyophilized and stored at −80 °C in a Labconco FreeZone 18 Liter Freeze Dry System (Labconco Corp., Kansas City, MO, USA). For the in vivo experiments, lyophilized extracts were reconstituted with 50 mL of distilled water. All samples were handled under dim-light conditions.

### 2.3. Phenolic Compound Extraction

In order to evaluate phenolics, flavonoids and antioxidant capacities, extractions were carried out with an aqueous solution of acidified methanol (50% *v/v*, pH 2.0) and aqueous acetone (70% *v/v*) prepared according to Osorio-Esquivel et al. [42] with modifications. A 0.25 g sample was shaken first with 10 mL of 50% (*v/v*) acidified (pH 2.0; HCl) methanol for 1 h at room temperature using an orbital shaker at 200 rpm. The mixture was centrifuged in an EBA 200 Hettich centrifuge (Andreas Hettich GmbH & Co. KG, Tuttlingen, Germany) at 1500× *g* for 10 min, and the supernatant was recovered (extract A). The insoluble residue was then re-extracted (extract B) with 10 mL of aqueous acetone (70% *v/v*) following the same procedure described above. Then, both supernatants were mixed (MAA extract) and stored at −20 °C until analysis. Determinations were performed in triplicate.

### 2.4. Determination of Total Soluble Phenols and Flavonoids

Total phenolics were measured according to the Folin-Ciocalteu method described by Osorio-Esquivel et al. [42] with slight modifications. The absorbance of the reaction was measured at 757 nm using a spectrophotometer (Biomate 3 Thermo Scientific, Waltham, MA, USA). A calibration curve was prepared under similar conditions using gallic acid as the standard, and results were expressed as milligrams of gallic acid equivalents per gram of dry sample. The total flavonoid determination was performed according to the aluminum chloride colorimetric method described by Osorio-Esquivel et al. [42]; the absorbance of the reaction was measured at 510 nm. Catechin was used as a standard for the calibration curve, and the results were expressed as milligrams of catechin equivalents per gram of dry sample.

### 2.5. Betalain Content

Betacyanin and betaxanthin content was analyzed following the method described by Sumaya-Martínez et al. [43] with slight modifications. Lyophilized extracts were reconstituted in distilled water at a 1:25 dilution and clarified at 12,000× *g* for 15 min at room temperature. For both pigments, determination was carried out photometrically and reported as milligrams of betanin/indicaxanthin equivalents per liter of clarified juice. Absorbances were measured at 535 nm for betacyanins and 484 nm for betaxanthins, and the content was calculated following the equation:Betacyanins or betaxanthins [mg/L] = [(A × DF × MW × 1000/ε × 1)]
where: A = absorbance at 535 or 484 nm, DF = dilution factor, MW = molecular weight (550 g/mol for betacyanins and 308 g/mol for betaxanthins), and ε = extinction coefficient (60,000 L/(mol cm) for betacyanins and 48,000 L/(mol cm) for betaxanthins).

### 2.6. Determination of Anti-Free Radical and Ferric Reducing Activities

The free radical-scavenging activity of MAA extract was determined using different methods. The DPPH (2,2-diphenyl-1-picrylhydrazyl) method was performed according to Yap et al. [44] with modifications. The ABTS^•+^ (2,2′-azino-bis-3-ethylbenzothiazoline-6-sulfonic acid) assay was performed following the method described by Ramlagan et al. [45] with modifications. The chelating activity was evaluated using the FRAP (ferric reducing antioxidant power) method as described by Tenore et al. and Yap et al. [44,46]. Antioxidant activities were expressed as micromoles of Trolox equivalents per gram of dry sample.

### 2.7. UPLC-QTOF-MS/MS Analysis

#### 2.7.1. Sample Preparation

Betanin standard (Sigma-Aldrich #901266-5G, St. Louis, MO, USA) was prepared at a concentration of 1 mg/mL in ultrapure water. After thawing at 4 °C, 0.11 g of lyophilized extract of OR was resuspended in 1 mL of water with 0.1% formic acid. The sample was vortexed for 60 s and centrifuged for 10 min at 186× *g*. It was then passed through a 0.22 µm nylon filter and centrifuged again for 5 min at 186× *g*. Finally, the supernatant was placed in an amber vial for analysis.

#### 2.7.2. Liquid Chromatography and Mass Spectrometry

Chromatographic separation was conducted on a Waters ACQUITY UPLC^®^ Class I (Waters Corporation, Milford, MA, USA) using an Acquity HSS T3 column (2.1 × 100 mm; 1.8 µm, Waters Corporation, Milford, MA USA). The conditions for UPLC were set as follows: column and autosampler were maintained at 30 °C and 6 °C, respectively, flow rate of 0.30 mL/min and the injection volume was 3 μL. The mobile phases consisted of (A) 0.1% formic acid in water and (B) 0.1% formic acid in acetonitrile. The separation of the compounds was in a linear gradient as follows: 1% B (0–3 min), 1–25% B (3–19 min), 25% B (19–20 min), 25–100% B (20–20.10 min), 100% B (20.10–21.10 min), 100–1% B (21.10–21.20 min) and 1% B (21.20–24 min).

In order to explore the main chemical constituents of OR extract, phytochemical profiling was performed. Mass spectrometry analysis was performed on a Waters Synapt G1 Q-TOF mass spectrometer (Waters Corporation, Milford, MA, USA) equipped with an electrospray ionization (ESI) source operating in both positive and negative ion mode. The MS conditions for the positive ion mode were set as follows: desolvation gas flow of 650 L/h at a temperature of 250 °C. The capillary voltage was 3.5 kV, and the sample and extraction cone voltages were 35 V and 4.5 V, respectively. Trap and transfer collision energies were 6 eV and 4 eV, respectively. For the negative ion mode the conditions were set as follows: desolvation gas flow of 500 L/h at a temperature of 300 °C. The capillary voltage was 2.3 kV, and the sample and extraction cone voltages were 35 V and 3.5 V, respectively. Trap and transfer collision energies were 4 eV and 2 eV, respectively. For both ion modes, the source temperature was 120 °C. The scan interval was 0.1 s, and the range of acquisition was 100–1000 *m/z*. For an accurate determination of mass, leucine-enkephalin was used as the lock mass, with an *m/z* value of 556.2771 for ESI (+) and an *m/z* value of 554.2615 for ESI (−).

Typical chromatograms were processed using the Progenesis QI software to obtain raw data, including precursor and fragment ions. Raw data were compared and classified according to online databases, including The Human Metabolome Database (https://hmdb.ca. accessed on 20 February 2022), ChemSpider (https://www.chemspider.com, accessed on 20 February 2022) and PubChem (https://pubchem.ncbi.nlm.nih.gov, Accessed on 20 February 2022). The compounds were classified according to their chemical structure, and their biological activity was analyzed according to the literature reviewed.

### 2.8. Animals

Sixty-four Wistar rats weighing 200–250 g were obtained from the animal facility at the Basic Sciences Center of the Autonomous University of Aguascalientes. Rats were housed in polypropylene cages at room temperature (25 ± 2 °C) under a daily 12 h light/day cycle and with free access to water and food. Experiments were performed following the guidelines of the local Ethics Committee from the Autonomous University of Aguascalientes (permission no. CEADI-UAA-02-2021), according to the Mexican governmental guideline NOM-062-ZOO-1999 and the guidelines of the National Institutes of Health for the care and use of Laboratory animals (NIH publications no. 8023).

### 2.9. Experimental Design

After two weeks of adaptation, animals were divided into 8 groups of 8 animals per group: Control group: untreated rats; DF group: rats who received a single dose of diclofenac (75 mg/kg, i.p.) [39]; OR group: rats who were prophylactically treated with *Opuntia robusta* fruit extract (800 mg/kg/5 days, orally) [17]; Bet group: rats who were prophylactically treated with betanin (25 mg/kg/5 days, orally) [47]; NAC group: rats who were prophylactically treated with *N*-acetylcysteine (50 mg/kg/5 days, i.p.) [48]. In the OR + DF, Bet + DF and NAC + DF groups, the antioxidants were administered for 5 days before the induction of acute toxicity with DF on the sixth day. Due to the wide range of doses reported in the literature, a dose–response curve was performed with male Wistar rats (results not shown). The betanin concentration in the administered dose of the OR extract (800 mg/kg) was 4.45 mg of betanin equivalents. Twenty-four hours after the last administration, rats were anesthetized with sodium pentobarbital (40 mg/kg) intraperitoneally, and transcardial perfusion was performed with wash solution (0.9% sodium chloride, 0.5% procaine and 0.1% heparin). Samples of liver tissue were collected, and biomarkers of oxidative stress, gene expression (*n* = 3) and protein expression (*n* = 4) were analyzed. For the histochemical study (*n* = 4), the above procedure was repeated followed by perfusion with 10% formaldehyde buffered to pH 7.0 with phosphate buffer.

### 2.10. Biomarkers of Oxidative Stress

For biochemical and molecular tests, different fragments of 1 cm^3^ were isolated and frozen at −80 °C. To determine the concentrations of reduced glutathione (GSH), a colorimetric assay kit was used (#38185, Sigma-Aldrich, St. Louis, MO, USA) following the manufacturer’s instructions. The final concentration of GSH was determined by the equation:GSH = (total glutathione − GSSG) × 2
and the ratio of GSSG to GSH was also determined. Malondialdehyde (MDA), a product of lipoperoxidation, was determined according to the method described by Uchiyama & Mihara [49] with modifications.

### 2.11. Molecular Biomarkers

#### 2.11.1. Quantitative RT-PCR Analysis

After perfusion, total RNA was isolated from 100 mg using the SV Total RNA Isolation System (Promega Corporation, Madison, WI, USA) according to the manufacturer’s protocol. The RNA was quantified using a Biodrop (Isogen Life Science, Barcelona, España) and stored at −80 °C until required. Reverse transcription was performed with 1 µg of total RNA using GoScript™ Reverse Transcriptase (Promega A5000, Madison, WI, USA) according to the manufacturer’s instructions. Subsequently, qPCR was performed using the Maxima SYBR Green/ROX qPCR Master Mix (2×) (K0221, Thermo Scientific, Waltham, MA, USA) using StepOneTM equipment (Applied Biosystems, Waltham, MA, USA) with the following thermocycling conditions: 50 °C for 2 min and 95 °C for 3 min, followed by 40 cycles of 95 °C for 45 s and 59 °C for 35 s and finally 95 °C for 15 s, 60 °C for 1 min and 95 °C for 15 s for the melt curve. The oligonucleotide primers are displayed in Appendix A. Relative expression levels were normalized to those of β-actin, and the differences were determined using the 2^–∆∆Ct^ method [50].

#### 2.11.2. Western Blot

For Western blotting, 40 µg of protein extract was separated on a 10% SDS-PAGE gel, and proteins were transferred to polyvinylidene difluoride (PVDF) membranes (Bio-Rad, Philadelphia, PA, USA). The membranes were blocked with Tris-buffered saline (TBS) and 5% skimmed milk for 1 h at room temperature. For immunodetection, the membranes were incubated overnight at 4 °C with the primary antibody, a Nrf2 polyclonal antibody (1:1000; Thermo Fisher Scientific, Waltham, MA, USA). Blots were incubated with goat anti-rabbit IgG conjugated HRP antibody (1:500; Thermo Fisher Scientific, Waltham, MA, USA). After the incubation, the membranes were washed with TBST (Tris-buffered saline-0.05% Tween 20), and blots were developed with Clarity Western ECL substrate (Bio-Rad, Hercules, CA, USA) for chemiluminescence imaging. The densitometric analysis of the obtained bands was performed using NIH ImageJ2 version 2.3.0/1.530 software. Intensity values were normalized to those of the internal housekeeping protein GAPDH and expressed as relative protein expression.

#### 2.11.3. Immunohistochemistry (IHC) for Active Caspase-3

To visualize the presence of active caspase-3 in liver tissue, sections were subjected to immunohistochemistry as described previously [51]. Samples were incubated with the primary antibody, a monoclonal antibody against active caspase-3 (1:200; Thermo Fisher Scientific, Waltham, MA, USA), overnight at 4 °C. The secondary antibody, goat anti-mouse HRP (1:100; Thermo Fisher Scientific, Waltham, MA, USA) was incubated for 2 h at room temperature. Slides were washed three times with PBS-Tween 20, and peroxidase activity was developed with diaminobenzidine (DAB) (Thermo Fisher, Waltham, MA, USA) for 5 min. Slides were again washed in 1× PBS and counterstained with hematoxylin (diluted 1:10 in tri-distilled water). For the negative control, the primary antibody was not added. Images were taken on a Leica ICC50W (Leica Biosystems, Buffalo Grove, IL, USA) microscope at 40× magnification and processed using Leica LAS EZ software. Quantification of active caspase-3 was performed by counting the number of positive cells (positive reaction in the cytoplasm) and reported as the mean number of positive cells per mm^2^. In addition, the reaction area intensity was evaluated based on optical density values of DAB and reported as the reaction area percentage. At least 18 fields per group (1.25 mm^2^) were evaluated.

### 2.12. Statistical Analysis

Statistical significance was evaluated by performing ANOVA 1-way (parametric) or Kruskal-Wallis (nonparametric) tests using GraphPad Prism 9 (GraphPad Software, San Diego, CA, USA). Normal distribution of the data was analyzed with the Shapiro-Wilk test.

For antioxidant analysis of the fruit extract, results are presented as the mean of each determination ± standard deviation (SD). In vivo results are presented as the mean of each group (n = 3 for oxidative stress markers and gene expression, n = 4 for Western blot and immunohistochemical analysis) ± standard error of the mean (SEM). *p* < 0.05 was considered statistically significant.

## 3. Results

### 3.1. Determination of Antioxidant Properties of O. robusta Fruit Extract

As shown in Table 1, the OR fruit extract showed a higher amount of total phenols (1330 ± 1.7 mg GAE/100 g dw) than flavonoids (1090 ± 0.9 mg CatE/100 g dw). With respect to the betalain content (641.1 ± 12.7), betacyanins showed a higher concentration (452.2 ± 9.0 mg BE/L) compared to that of betaxanthins (188.9 ± 3.7 IxE/L).

The capacity of OR extract to scavenge the radicals DPPH and ABTS^•+^ was 30.9 ± 1.3 and 102.6 ± 5.2 µmol TE/g dw, respectively, and the capacity to reduce ferric ions was 95.8 ± 7.3 µmol TE/g dw according to the FRAP assays (Table 2).

### 3.2. Identification of the Main Bioactive Compounds in O. robusta Fruit Extract 

Figure 1A shows the mass spectrum of betanin with a [M+H]+ peak at *m/z* 551.1998 and a retention time of 9.10 min, as well as the extracted ion chromatogram (EIC). For the MS/MS analysis, the precursor of betanin molecular ion [M+H]+ was filtered to obtain the spectrum and a molecular ion of *m/z* 389.1379. Figure 1B shows the mass spectrum of the OR fruit extract with the same fragmentation pattern as that of the betanin standard. This result confirms the presence of betanin in the OR fruit extract.

In total, 50 tentative compounds, including 15 organo-oxygen compounds, 9 betalains and 26 other compounds, were determined in the OR fruit extract. In Table 3, 12 compounds are listed for the positive ion mode, including the presence of vitamin C (*m/z* 11.0387 and retention time of 5.197 min) and betalains: betalamic acid (*m/z* 212.055), indicaxanthin (*m/z* 309.0984), neobetanin (*m/z* 549.1382), gomphrenin-I (*m/z* 551.1633) and betanin (*m/z* 551.1498). In Table 4, 22 compounds are listed for the negative ion mode, including phenols, such as 5-hydroxyconiferyl alcohol (*m/z* 195.382), flavonoids, such as hesperetin 5-*O*-glucoside (*m/z* 463.1337), and betalains, such as vulgaxanthin I (*m/z* 384.1014). 

Other compounds with possible biological activity are listed in Appendix A.

### 3.3. Biomarkers of Oxidative Stress

Figure 2 shows the oxidative stress biomarkers present in tissue homogenates. There was a significant increase of 50.85% in MDA levels in the group treated with DF (59.92 ± 5.33 nmol/100 mg) compared to those in the control group (39.72 ± 6.75 nmol/100 mg) (*p* < 0.05) (Figure 2A). Pretreatment with OR fruit extract, Bet and NAC followed by DF administration significantly reduced MDA levels (19.43% for OR + DF, 23.79% for Bet + DF and 17.84% for NAC + DF) compared to those with DF alone (*p* < 0.05).

DF administration induced a significant decrease in GSH concentrations (Figure 2B) of 65% (62.90 ± 0.42 μmol/L) compared to those in the control group (96.21 ± 3.31 μmol/L) (*p* < 0.05). The prophylactic administration of OR, Bet and NAC maintained GSH concentrations at levels comparable to those in the control group with no statistical difference, but concentrations were significantly (*p* < 0.05) higher (40.23, 124.2 and 44.25% respectively) when they were administered to the DF groups compared to those with DF alone. We also measured the ratio of GSSG to reduced GSH in the liver after DF challenge (Figure 2B), and there was a significant decrease in the DF group pretreated with Bet (141.03 ± 8.63 μmol/L) compared to that in the DF group.

### 3.4. Relative Expression of Genes Related to the Constitutive and Inducible Antioxidant Response

The relative gene expression of catalase was significantly decreased, compared to that in the control group, in the DF (0.8-fold; *p* < 0.0001), OR + DF (0.8-fold; *p* < 0.001), Bet + DF (0.9-fold; *p* < 0.0001), NAC + DF (0.7-fold; *p* < 0.01) and NAC (0.3-fold; *p* < 0.01) groups (Figure 3A). Sod1 showed a significant increase in its expression only in the OR group (0.9-fold higher; *p* < 0.001: Control vs. OR) (Figure 3B). *Sod2* expression was increased in the DF (6.3-fold; *p* < 0.01: Control vs. DF), OR + DF (10.4-fold; *p* < 0.0001: Control vs. OR + DF) and Bet + DF (6.9-fold; *p* < 0.0001: Control vs. Bet + DF) groups (Figure 3C). *Nrf2* expression was decreased in the DF (0.3-fold; *p* < 0.0001: Control vs. DF), OR + DF (0.6-fold; *p* < 0.05: Control vs. OR + DF), Bet + DF (0.7-fold; *p* < 0.0001: Control vs. Bet + DF), NAC + DF (0.6-fold; *p* < 0.05: Control vs. NAC + DF) and NAC (0.4-fold; *p* < 0.01: Control vs. NAC) groups (Figure 3D). Heme-oxygenase 1 (Hmox1) expression was significantly induced in the DF (24.9-fold; *p* < 0.0001: Control vs. DF), OR + DF (15.7-fold; *p* < 0.0001: Control vs. OR + DF) and Bet + DF (4.7-fold; *p* < 0.05: Control vs. Bet + DF) groups (Figure 3E). The expression of NAD(P)H dehydrogenase [quinone] 1 (Nqo1) was significantly increased in the DF (2.7-fold; *p* < 0.001: Control vs. DF), OR + DF (5.5-fold; *p* < 0.0001: Control vs. OR + DF), Bet + DF (2-fold; *p* < 0.01: Control vs. Bet + DF) and Bet (2-fold; *p* < 0.05: Control vs. Bet) groups (Figure 3F). The expression of glutamate-cysteine ligase catalytic subunit (Gclc) was significantly increased only in the group treated with DF (1.3-fold; *p* < 0.001: Control vs. DF). Finally, expression of the DNA damage-inducible growth arrest gene (Gadd45a) was significantly increased in the DF (2.3-fold; *p* < 0.001: Control vs. DF) and OR + DF (2.9-fold; *p* < 0.0001: Control vs. OR + DF) groups. No significant differences were observed in the relative expression of caspase-3 (Casp3 gene) (Figure 3I).

### 3.5. Nrf2 Protein Expression

Figure 4A shows the Western blot for the Nrf2 protein in liver tissue. Densitometric analysis (Figure 4B) showed a significant increase in Nrf2 protein expression in the DF (1.6-fold; *p* < 0.0001), OR + DF (3-fold; *p* < 0.0001), Bet + DF (3.5-fold; *p* < 0.0001), NAC + DF (2.7-fold; *p* < 0.0001), Bet (0.9-fold; *p* < 0.05) and NAC (1.2-fold; *p* < 0.001) group, compared to that in the control group. It should be noted that the highest expression was observed in the OR + DF (1.4-fold, *p* < 0.001), Bet + DF (1.8-fold, *p* < 0.0001) and NAC + DF (1.1-fold, *p* < 0.05) groups when compared to that in the DF group.

### 3.6. Apoptosis and Active Caspase-3 Evaluation

Immunohistochemical analysis showed a significant increase in the number of active caspase-3-positive cells (641%) in the group treated with diclofenac, equivalent to 441 cells on average/mm^2^ (*p* < 0.0001) (Figure 5A,B), with an increase in reaction intensity of 20.15% (*p* < 0.0001) (Figure 5C) with respect to that in the control group (69 cells/mm^2^). The groups treated prophylactically with OR fruit extract, Bet and NAC showed a number of active caspase-3 positive cells similar to that in the control group, with mean values of 48, 60 and 36 (cells/mm^2^), respectively.

## 4. Discussion

Phytochemicals are bioactive compounds that have beneficial effects on health. Their biological activity depends on their chemical structure, and they occur in a variety of foods. The nutraceutical value of a food is based on the amount and type of phytochemicals it contains [52]. It has been reported that OR pulp contains antioxidants, such as phenolics, flavonoids, ascorbic acid (vitamin C) and betalains [17,38,43].

In our study, the amount of phenolic compounds obtained (1330 ± 1.7 mg GAE/100 g dw) in the OR fruit extract was similar to that reported by pulido-Hornedo et al. [7] and higher than that in *O. ficus-indica* (89.2 ± 3.6 mg GAE/100 g) [53] and *O. streptacantha* (104.66 ± 1.51 mg GAE/100 g) [54]. Phenolic compounds are phytochemicals that have an aromatic ring with at least one hydroxyl substituent (phenol), which gives them important antioxidant characteristics. The three main mechanisms of action are derived from their direct reaction with free radicals: (1) the transfer of a hydrogen atom (HAT), (2) single electron transfer (SET), and (3) chelation of free metal ions, such as Fe(III). Based on these interactions, an antioxidant can act as a scavenger when the final product generates another less reactive radical or as a quencher when it completely neutralizes the radical, generating stable molecules [55].

Flavonoids are phenols widely distributed in plant foods, and their concentration (1090 ± 0.9 mg CatE/100 g dw) in the OR fruit extract was higher than that reported for OR pulp (793 mg CAE/100 g dw) [7] or *O. ficus-indica* (red cultivar, 35.19 ± 2.08 mg CATE/100 g dw) [56]. The basic structure of these compounds is the flavan nucleus (2-phenyl-benzo-c-pyran) and a system of two benzene rings (A and B) linked by an oxygen-containing pyran ring (C), which gives rise to the subclasses: flavones, isoflavonones, flavanols, flavonols and flavanones. These phytochemicals can inhibit the enzyme nitric oxide synthase (NOS) that generates the nitric oxide radical (^•^NO), preventing the formation of radicals, such as peroxynitrite, which is highly reactive with essential macromolecules [57].

The high concentrations of betacyanins (452.2 ± 9.0 mg BE/L) and betaxanthins (188.90 ± 3.69 mg IxE/L) in the OR fruit extract coincide with previous reports [7,17]. Betalains are a group of pigments that occur in 13 families of Caryophyllales, including the Cactaceae family [58]. These immonium derivatives of betalamic acid are divided intro betacyanins, responsible for the red–purple coloration (λ ≈ 535 nm), and betaxanthins, responsible for the yellow coloration (λ ≈ 480 nm) [59]. These pigments are more abundant in the fruits with red–purple colorations. Our results confirm that the OR fruit has the highest concentrations of betalains, mainly betacyanins, within the Cacti family [17]. Czapski et al. [60] suggest a strong correlation between antioxidant capacity and red pigment content; therefore, OR fruit represents a rich source of antioxidant pigments.

Since the initial analysis of phytochemicals showed that the OR extract is a rich source of betalains, molecular analysis was performed via UPLC-QTOF-MS/MS. Betalamic acid was identified, and this compound has been reported in *O. ficus-indica* [61] and *O. stricta* [8]; its antioxidant activity is due to the fact that it reduces two Fe(III) ions into Fe(II) [62], preventing the Fenton reaction and the generation of the hydroxyl radical (^•^OH). Two betaxanthins were also identified: indicaxanthin and vulgaxanthin I. These pigments are the product of the condensation of betalamic acid with amines or their derivatives [58], and it has been suggested that their antioxidant activity is due the presence of 2-3 imino groups (=NH) and 1-2 hydroxyl groups (-OH) in the amines or amino acid moiety [63].

Indicaxanthin is found in large quantities in *O. ficus-indica* [64], and in addition to its antioxidant activity, it has been shown in in vitro studies: (1) to modulate the expression of the intercellular adhesion molecule ICAM-1 in human umbilical vein endothelial cells [65]; (2) to prevent atherogenic formation by inhibiting the overexpression of NADPH oxidase 4 (NOX-4); (3) to inhibit the activation of nuclear enhancing factor of NF-κB; and (4) to prevent the apoptosis of cells of a human monocytic cell line (THP-1) [40]. Vulgaxanthin I has been found in *O. ficus-indica* and *Beta vulgaris* (yellow beet) and in Amaranthaceae [21,63,66].

Betacyanins are divided into betanin-type, gomphrenin-type and amaranthine-type. Gomphrenin-I has been reported in species, such as *Gomphrena globosa* [67], and was reported for the first time in *O. robusta* by Stintzing et al. (2005) [66]. It has a chemical structure similar to that of betanin, and it has been suggested that its high antioxidant capacity is due to the presence of the -OH group at carbon 5 of betanidin (general structure of betacyanins), unlike betanin, which presents it at C-6 [63,68]. Likewise, the UPLC-QTOF-MS/MS analysis showed the presence of betanin in OR fruit extract, consistent with previous reports [17,21,69]. Other studies show the same fragmentation pattern (parent ion *m/z* 551.19 and daughter ion *m/z* 389.13) for species such as *B. vulgaris* [70], *Hylocereus polyrhizus* [71] and *O ficus-indica* [61]. The antioxidant effects of betanin have been widely described: the inhibition of lipoperoxidation and LDL [64,72,73,74], as well as the increase in GSH synthesis [59]. It has been reported that the antioxidant activity of betanin is due to its capacity to donate electrons and hydrogen atoms, which comes from the phenolic hydroxyl group attached to C-6. The resulting product is a betanin radical that can be broken down into betalamic acid and the stable cyclo-DOPA 5-O-B-glucoside radical [75].

The presence of vitamin C is in line with what has been reported previously for OR [38]. Recently, Pulido-Hornedo et al. [7] quantified vitamin C (141.14 mg/100 g) in the OR fruit extract via HPLC-UV. This vitamin is part of a series of redox cycles in which non-enzymatic reducers (vitamin E, glutathione and NADPH) and enzymatic reducers (glutaredoxin and glutathione reductase) participate, stopping the propagation of peroxidative processes and repairing peroxidized lipids in biological membranes [76,77]. On the other hand, it neutralizes ROS, such as superoxide anion and hydroxyl radical [78].

Phenolic acids are phenols that have a carboxylic acid functionality that depending on the constitutive carbon framework they contain are divided into hydroxycinnamic acids and hydroxybenzoic acids [79]. We identified 3,4-*O*-dimethylgallic acid (*m/z* 216.08) and cinnamic acid (*m/z* 131.04), 2-*O*-galloyl-1,4-galactarolactone (*m/z* 380.98) and vanillic acid (*m/z* 167.03), which are compounds for which the antioxidant capacity is due to the number and position of the phenolic hydroxyl groups, in addition to the methoxy and carboxylic acid groups [80]. Likewise, 5-hydroxyconiferyl alcohol was identified, which is an intermediate in the synthesis of lignans; its antioxidant activity has been described, although little has been explored [81,82,83]. The flavonoid identified, hesperetin 5-*O*-glucoside (*m/z* 463.13), has antidiabetic activity owing to its glycosylated portion at the C-5 position [84].

Antioxidant activity is one of the most described biological effects of phytochemicals, determined using in vitro assays where the ability to quench radicals, such as ABTS^•+^ and DPPH, is measured [63,64], as well as reducing ferric ions [85]. In the present study, the OR fruit extract showed a higher antioxidant capacity (102.6 ± 5.2 μmol TE/g dw) in the ABTS^•+^ assay compared to that reported for the OR pulp (62.2 ± 5.0 μmol TE/g dw) [7] and the Spanish prickly pear *O. ficus-indica*, (6.70 ± 0.73 μmol TE/g) [86]. Likewise, the value obtained (30.09 ± 1.4 μmol TE/g dw) in the DPPH assay was higher than that reported for the Spanish prickly pear (5.22 ± 0.89 μmol TE/g) [86], indicating that the phytochemicals in the OR fruit extract are good electron and hydrogen atom donors. The FRAP assay value of the OR fruit extract (95.8 ± 7.3 μmol TE/g dw) was higher than that of the OR pulp (62.2 ± 5.0 μmol TE/g) [7] and less than that of the ethyl ethanolic extract of blueberry (*Rubus* spp.): 258.9 ± 11.69 μmol TE/g [87].

Our results suggest through the different antioxidant mechanisms described that the beneficial effects of the OR fruit extract can be attributed to the biological diversity of phytochemicals present in the fruit, apart from betanin, such as gomphrenin I, vulgaxanthin I, indicaxanthin and phenolic acids, as well as other important bioactive compounds that have not been studied in detail (Appendix A).

In our in vivo model, the OR fruit extract protected the liver from damage caused by a high dose of diclofenac (DF). The hydroxylation of DF, mediated by the cytochrome P450 enzymes CYP2C9 and CYP3A4 [34,39], generates the metabolites 4′-OH-DF and 5-OH-DF, respectively, which can be oxidized to form the electrophilic intermediates DF-1,′4′-iminoquinone and DF-2,5-iminoquinone [37] and induces redox cycles that increase ROS, RNS generation and cellular oxidative stress [35,36,37]. ROS (e.g., O_2_^•−^, ^•^OH) can initiate lipid peroxidation that results in the formation of lipid aldehydes, such as malondialdehyde (MDA) [88]. In our model, prophylactic administration with the antioxidants OR, Bet and NAC decreased the concentration of MDA to a level comparable to that in the control group, indicating that they prevent lipid peroxidation caused by the reactive metabolites derived from DF. In another study, a decrease in hepatic MDA levels was also reported after pretreatment with OR in a model of acute hepatotoxicity induced by APAP [17].

Reduced glutathione (GSH) is a non-enzymatic endogenous antioxidant that maintains the redox balance [89], and its oxidized form (GSSG) increases under oxidative stress. The bioactive compounds of the OR fruit extract can directly interact with GSSG and reduce it to GSH. Esatbeyoglu et al. [90] suggested that betanin induces GSH synthesis via the Nrf2 pathway, which explains the high hepatic concentrations of GSH in the group pretreated with betanin. GSH is an important antioxidant since it is also a substrate for glutathione peroxidase (GPx), which reduces lipid hydroperoxide radicals into hydroxy-fatty acids. Likewise, this reaction causes the reduction of vitamin C, which in turn reduces vitamin E [76,91].

The antioxidant enzymes CAT, SOD1 and SOD2 are constitutively expressed in peroxisomes, the cytoplasm and mitochondria, respectively. In our study, the increase in gene expression of the enzyme *Sod2* in the group treated with DF shows that this drug induces mitochondrial injury by increasing the generation of superoxide anions [35,37]. We show that the OR fruit extract protects against this oxidative damage by increasing the expression of *Sod2*, in line with previous in vitro studies [39]. *Sod2* gene expression is regulated by transcription factors such as NF-κB and Nrf2, important regulators of the antioxidant response. In basal conditions, Nrf2 is inactive in the cytosol and bound to Keap1. Nrf2 activators cause its uncoupling from Keap1 via different pathways, the most important one being the oxidation of Keap1 cysteine residues mediated by electrophilic compounds and other oxidants [92]. Although a decrease in *Nrf2* gene expression was observed, our Western blot results show its post-translational activation. Once uncoupled from Keap1, Nrf2 translocates to the nucleus where it heterodimerizes with sMaf proteins and binds to electrophilic response gene (EpRE) motifs that encode detoxifying enzymes (e.g., *Nqo1*), antioxidant enzymes (e.g., *Sod2*, *Cat*), heme detoxifying enzymes (e.g., *Hmox1*) and enzymes involved in GSH synthesis (e.g., *Gclc*) [93,94].

Bioactive compounds are also capable of activating and/or enhancing cellular antioxidant responses. It has been reported that betanin induces the antioxidant response mediated by the Nrf2 pathway and its downstream proteins NQO1 (*Nqo1* gene) and HO-1 (*Hmox1* gene) in HepG2 hepatoma cells [90], as well as the induction of phase II detoxifying enzymes, such as glutathione S-transferase P (GSTP), glutathione S-transferase mu (GSTM) and NQO1, via the activation of mitogen-activated protein kinases [41]. In another study, polysaccharides, extracted from the cladode of *O. milpa alta*, protected pancreatic β-cells against alloxan-induced apoptosis and oxidative stress via the Nrf2-mediated induction of γ-glutamyl-cysteine synthetase (γ-GCSc) [95]. We have previously evaluated the effect of OR fruit extract on *Sod2*, *Hmox1* and *Gclc* gene expression in an in vivo model of APA-induced acute hepatotoxicity [17]. In this study, we observed that pretreatment with OR fruit extract increased the expression of the *Sod2*, *Hmox1* and *Nqo1* genes via the post-translational activation of Nrf2. As expected, acute DF toxicity induced the activation of Nrf2; however, the protein levels of Nrf2 were higher in the groups subjected to prophylactic treatments with OR, Bet and NAC, with Bet being the antioxidant that induced the highest expression levels of Nrf2.

*Hmox1* is a downstream target gene of Nrf2. It encodes a protein that catalyzes the oxidative degradation of the heme group, resulting in the generation of anti-inflammatory and cytoprotective products: bilirubin, biliverdin and CO [90,96]. Under stress conditions, the induction of *Hmox1* expression is modulated by intracellular GSH depletion [97,98,99]. Our results showed an increase in the expression of *Hmox1* in the DF group and a significant diminution after pretreatments with OR, Bet and NAC, similar to that reported with APAP-induced liver injury [17]. As previously described, GSH levels decreased in the group treated with DF, while they were restored in the groups that received the prophylactic treatments (OR + DF, Bet + DF, NAC + DF).

In our study, the prophylactic administration of OR fruit extract induced high expression of *Nqo1*, indicating that iminoquinones are one of the major metabolites generated from DF and that OR fruit extract can detoxify these metabolites. Iminoquinones are capable of damaging proteins and other biomolecules. The protein that *Nqo1* encodes is responsible for catalyzing the reduction of quinones to hydroquinones, facilitating their excretion [41].

In order to elucidate whether antioxidants present in OR fruit extract induce GSH synthesis, the expression of *Gclc*, the gene that encodes the enzyme that is important for GSH synthesis, was determined [100]. No significant changes in the expression of this gene were observed. These results indicate that the OR fruit extract has different mechanisms of action and that the biocomponents that it contains can act directly on the reactive chemical species and/or directly restore GSSG to its reduced state (GSH). On the other hand, since the prophylactic administration of Bet resulted in a significant increase in hepatic GSH concentrations, it is likely that this component can also induce its synthesis [90].

We recently reported that OR fruit extract protects against apoptotic cell death in a model of hepatocyte injury [39]. Our present results are in line with this cytoprotective effect. Although no differences were observed in the gene expression of caspase-3 in the experimental groups, our immunohistochemical analysis showed a significant increase in the number of cells positive for active caspase-3 in the group treated with DF. In normal conditions, caspase proteins are present as inactive procaspases [101]. The increase in the number of caspase-3-positive cells demonstrates increased apoptotic cell death in the DF toxicity model. DNA damage can activate apoptosis, which can be caused by the peroxynitrite radical (ONOO-) [102].

Ferroptosis is a type of cell death related to an increase in Fe^2+^, the Fenton reaction that produces the ^•^OH radical, lipid peroxidation, GSH depletion and induction of the Nrf2-mediated antioxidant response [103,104,105,106,107,108]. Our results suggest that the OR extract may have a protective effect against this type of death due to the chelating power of iron, the decrease in MDA, the restoration of GSG and the increase in the expression of Nrf2 and Nqo1 [107,108].

Cells have DNA damage-sensing mechanisms. One of these is the DNA damage-inducible growth arrest gene (*Gadd45a*), which was induced by the prophylactic administration of OR fruit extract. *Gadd45a* expression is rapid and dose-dependent, and it has been reported to be involved in DNA repair [109,110]. This gene can be induced via different pathways, one of them being mediated by p53. Since we previously reported that OR fruit extract downregulates the expression of p53 in vitro [39], it is suggested that in the present study, the regulation of GADD45a by OR fruit extract is mediated by a p53-independent pathway, e.g., via the MAP kinase pathway [110].

According to our results, the mechanisms underlying OR extract activity that provide hepatoprotection against oxidative stress are as follows: (1) chelation of iron and thus prevention of the Fenton reaction and lipid peroxide formation, (2) promotion of the reduction of peroxidized lipids (L-OOH) to lipid alcohols (L-OH) by restoring GSH levels, (3) direct interactions with ROS as scavengers or quenchers, (4) induction of the expression of antioxidant response genes regulated by NRf2, and (5) prevention of cell death mediated by apoptosis and possibly by ferroptosis.

Our study did not address iDILI caused by DF, and it did not explore risk factors that increase the development of liver damage due to DF consumption, due to age and the presence of previous pathologies, such as diabetes and hypertension present in a significant sector of the population older than 60 years, in which there is an increase in the incidence of liver damage caused by DF. However, cellular oxidative stress is a key factor involved in the pathogenesis of DILI, which was addressed in this study.

## 5. Conclusions

We observed that OR fruit extract is a rich source of beneficial, health promoting biological compounds with strong antioxidant and cytoprotective effects against mitochondrial damage and oxidative stress generated by DF. Both Bet and OR fruit extract exert their effects via a direct interaction with reactive species, the restoration of GSH levels and indirect effects via the regulation of genes involved in the antioxidant response mediated by Nrf2. The protective effects of the OR fruit extract and betanin are similar to those of NAC, which is currently used to treat acute liver injury, suggesting its potential use in clinical practice as a complement or alternative to treat acute liver injury related to cellular oxidative stress.

## Figures and Tables

**Figure 1 antioxidants-12-00113-f001:**
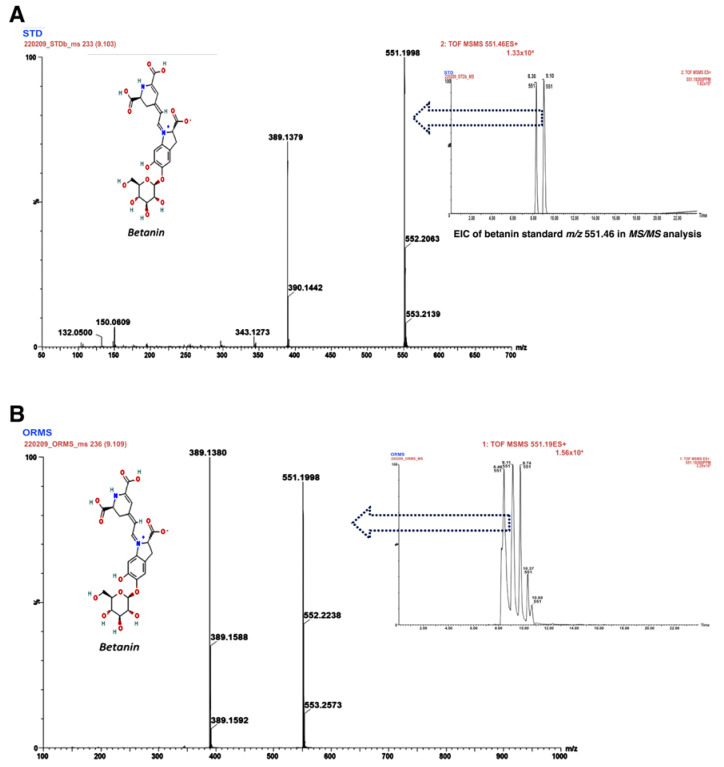
Mass spectra of betanin standard and *Opuntia robusta* extract. (**A**) Extracted ion chromatogram (EIC) and mass spectrum of betanin standard. (**B**) Mass spectrum of *O. robusta* sample. The EIC and mass spectra were analyzed via UPLC–QTOF–MS/MS.

**Figure 2 antioxidants-12-00113-f002:**
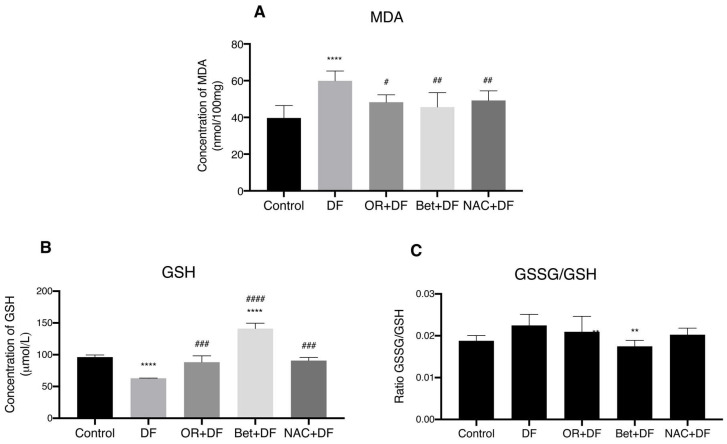
Oxidative stress biomarkers in livers of rats that received DF treatment and pretreatments with *O. robusta* extract (OR + DF), betanin (Bet + DF) and NAC (NAC + DF). (**A**) MDA concentrations, (**B**) GSH levels and (**C**) GSSG/GSH ratio. Bar graphs show the mean (*n* = 3) ± SEM; *p* > 0.05, **: *p* ≤ 0.01, ****: *p* ≤ 0.0001 vs. control group. #: *p* ≤ 0.05, ##: *p* ≤ 0.01, ###: *p* ≤ 0.001, ####: *p* ≤ 0.0001 vs. DF group.

**Figure 3 antioxidants-12-00113-f003:**
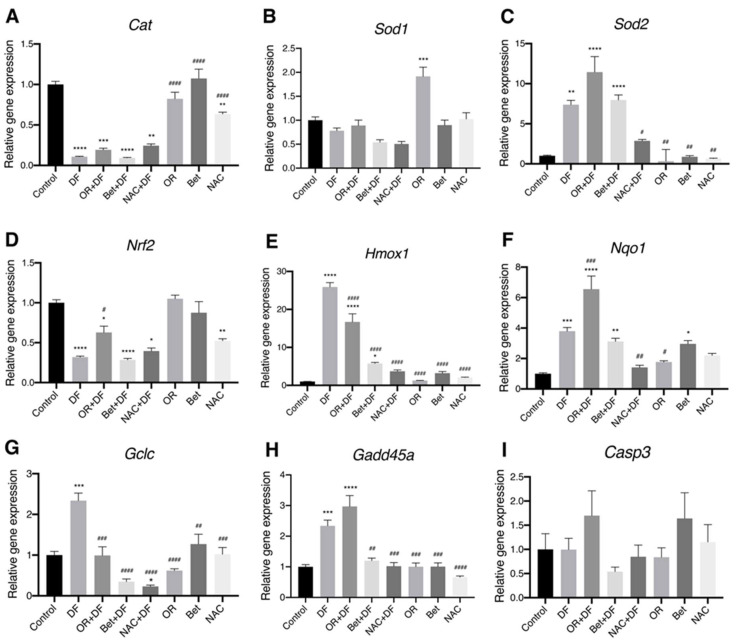
Relative gene expression of antioxidant genes: (**A**) *Cat*, (**B**) *Sod1*, (**C**) *Sod2*, detoxifying genes: (**D**) *Nrf2*, (**E**) *Hmox1*, (**F**) *Nqo1* and (**G**) *Gclc*, the DNA damage-inducible gene: (**H**) *Gadd45a* and the cell death gene (**I**): Casp3 after DF treatment and OR, Bet and NAC treatments and pretreatments in rat livers. Bar graphs show the mean (n = 3) ± SEM; *p* > 0.05, *: *p* ≤ 0.05, **: *p* ≤ 0.01, ***: *p* ≤ 0.001, ****: *p* ≤ 0.0001 vs. Control group. #: *p* ≤ 0.05, ##: *p* ≤ 0.01, ###: *p* ≤ 0.001, ####: *p* ≤ 0.0001 vs. DF group.

**Figure 4 antioxidants-12-00113-f004:**
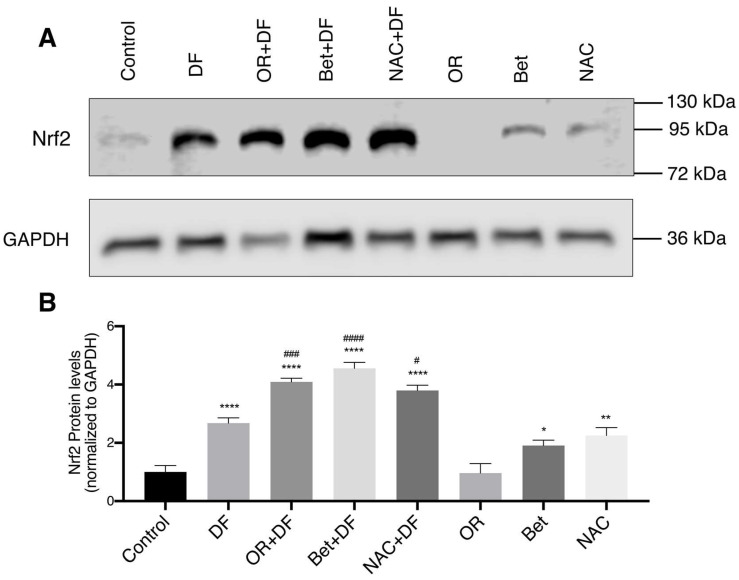
Western blot analysis and quantification of Nrf2 expression in rat liver after DF treatment and OR, Bet and NAC treatments and pretreatments in rat livers. (**A**) Nrf2 and GAPDH blots; (**B**) graphs showing the relative Nrf2 protein levels normalized to GAPDH from triplicate samples. Bar graphs show the mean (n = 4) ± SEM; *p* > 0.05, *: *p* ≤ 0.05, **: *p* ≤ 0.01, ****: *p* ≤ 0.0001 vs. Control group. #: *p* ≤ 0.05, ###: *p* ≤ 0.001, ####: *p* ≤ 0.0001 vs. DF group.

**Figure 5 antioxidants-12-00113-f005:**
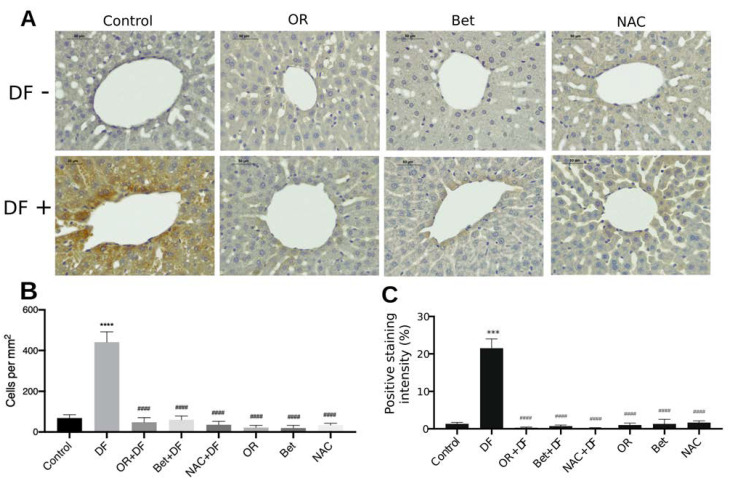
Immunohistochemical staining and quantification of active caspase-3 in rat liver after DF treatment and OR, Bet and NAC treatments and pretreatments in rat livers. (**A**) Representative tissue sections are shown. 400×. Scale bar = 30 µm. (**B**,**C**) Photographs of 18–22 fields (1.25 mm^2^) per group were taken, and the number of positive hepatocytes was determined, in addition to the intensity of the reaction area, using NIH ImageJ ver 2.3.0/1.530 software. Bar graphs show the mean (n = 4) ± SEM. Control group. *p* > 0.05, ***: *p* ≤ 0.001, ****: *p* ≤ 0.0001 vs. Control group. ####: *p* ≤ 0.0001 vs. DF group.

**Table 1 antioxidants-12-00113-t001:** Total phenols, flavonoids, betacyanins, betaxanthins and total betalain values in *O. robusta* extract.

MMA Extraction		H_2_O Extraction		
Total Phenols (mg GAE/100 g dw)	Total Flavonoids (mg CatE/100 g dw)	Betacyanins (mg BE/L)	Betaxanthins (mg IxE/L)	Total Betalains (mg betalains/L)
1330 ± 1.7	1090 ± 0.9	452.2 ± 9.0	188.9 ± 3.7	641.1 ± 12.7

Values represent the mean ± SD of three measurements; GAE—Gallic Acid Equivalents; CatE—Catechin Equivalents; BE—Betacyanin Equivalents; IxE—Indicaxanthin Equivalents; dw—dry weight.

**Table 2 antioxidants-12-00113-t002:** Antioxidant activity of *O. robusta* extract determined based on the DPPH, ABTS^●+^ and FRAP assays.

Antioxidant Activity of MAA Extract (μmol TE/g dw)		
DPPH	ABTS^●+^	FRAP
30.9 ± 1.3	102.6 ± 5.2	95.8 ± 7.3

Values represent the mean ± SD of three measurements; TE—Trolox Equivalents; dw—dry weight.

**Table 3 antioxidants-12-00113-t003:** Compounds with biological activity identified in *O. robusta* extract in positive ion mode.

Compound	Formula	Adduct	*m/z*	Retention Time (min)
Oxanes				
1,5-Anhydro-D-fructose	C_6_H_10_O_5_	M+H-H_2_O	145.0488	0.919
Cinnamic acids and derivatives				
Cinnamic acid	C_9_H_8_O_2_	M+H-H_2_O	131.0495	14.037
Phenylpropanoic acids				
Hydrocinnamic acid	C_9_H_10_O_2_	M+H-H_2_O	133.0641	16.460
Benzene and derivatives				
3,4-*O*-Dimethylgallic acid	C_9_H_10_O_5_	M+NH_4_	216.0867	5.119
Indanes				
1-Indanone	C_9_H_8_O	M+H	133.0634	11.692
Vitamins				
Vitamin C	C_6_H_8_O_6_	M+H	177.0397	5.197
Carboxylic acids and derivatives				
Betaine	C_5_H_11_NO_2_	M+H	118.0821	0.919
Betalains				
Betalamic acid	C_9_H_9_NO_5_	M+H	212.055	1.048
Indicaxanthin	C_14_H_16_N_2_O_6_	M+H	309.0984	8.830
Neobetanin	C_24_H_24_N_2_O_13_	M+H	549.1382	11.613
Gomphrenin-I	C_24_H_26_N_2_O_13_	M+H	551.1633	10.764
Betanin	C_24_H_26_N_2_O_13_	M+H	551.1498	9.887

**Table 4 antioxidants-12-00113-t004:** Compounds with biological activity identified in *O. robusta* extract in negative ion mode.

Compound	Formula	Adduct	*m/z*	Retention Time (min)
Lactones				
D-Glucaro-1,4-lactone	C_6_H_8_O_7_	M-H	191.0223	1.616
Cinnamic acids and derivatives				
1-*O*-Sinapoylglucose	C_17_H_22_O_10_	M-H	385.1205	11.743
Benzene and derivatives				
Vanillic acid	C_8_H_8_O_4_	M-H	167.0382	7.284
2-*O*-Galloyl-1,4-galactarolactone	C_13_H_12_O_11_	M+K-2H	380.9824	21.898
Carboxylic acids and derivatives				
2-*O*-Caffeoylhydroxycitric acid	C_15_H_14_O_11_	M+Na-2H	391.0282	1.486
Organooxygen compounds				
Cis-5-Caffeoylquinic acid	C_16_H_18_O_9_	M+K-2H	391.0376	1.099
trans-o-Coumaric acid 2-glucoside	C_15_H_18_O_8_	M+H-H_2_O	309.0968	6.176
trans-p-Coumaric acid 4-glucoside	C_15_H_18_O_8_	M+H-H_2_O	309.0969	6.176
6-Caffeoylsucrose	C_21_H_28_O_14_	M-H	503.138	7.902
Gentiobiosyl 2-methyl-6-oxo-2E,4E-heptadienoate	C_20_H_30_O_13_	M-H	477.159	9.398
Glucocaffeic acid	C_15_H_18_O_9_	M-H	341.0926	10.455
Coumarins and derivatives				
Rutaretin 9-rutinoside	C_26_H_34_O_14_	M+FA-H	615.202	6.665
Fatty acyls				
1-Hexanol arabinosylglucoside	C_17_H_32_O_10_	M+Na-2H	417.171	7.902
Flavonoids				
Hesperetin 5-*O*-glucoside	C_22_H_24_O_11_	M-H	463.1337	9.038
Phenol lipids				
Caryoptosidic acid	C_16_H_24_O_11_	M-H_2_O-H	373.119	9.578
Hydroxyisonobilin	C_20_H_26_O_6_	M-H	361.1717	17.079
Furanoid lignans				
Divanillyltetrahydrofuran ferulate	C_30_H_32_O_8_	M+FA-H	565.197	16.022
Phenols				
5-Hydroxyconiferyl alcohol	C_10_H_12_O_4_	M-H	195.0701	16.382
Betalains				
Betalamic acid	C_9_H_9_NO_5_	2M-H	211.0525	4.321
Vulgaxanthin I	C_14_H_17_N_3_O_7_	M+FA-H	384.1014	6.744
Betanin	C_24_H_26_N_2_O_13_	2M-H	1099.287	7.852
Betalamic acid	C_9_H_9_NO_5_	M-H_2_O-H	192.0344	8.909

## Data Availability

All of the data is contained within the article and the Appendix A.

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
