# Peer review of "Molecular and Antioxidant Characterization of Opuntia robusta Fruit Extract and Its Protective Effect against Diclofenac-Induced Acute Liver Injury in an In Vivo Rat Model"

_antioxidants, 2023, doi:10.3390/antiox12010113_

Round 1

Reviewer 1 Report

This work by Villa-Jaimes et al. aimed to investigate the phytochemical fingerprint of an Opuntia robusta fruit extract and its protective effect against diclofenac-induced acute liver injury.

The paper is adequately written, the introduction provides the reader with most of the state-of-the art in the research field, materials are adequately described, results well presented and discussed.

The following minor points should be addressed before the manuscript can be accepted for publication in Antioxidants.

-       Author must justify the dose of betanin administered and its relationship with the dose of OR

-       Which is the concentration of Betanin in the OR extract administered?

-       Authors often refer to indicaxanthin in the paper notwithstanding they administered betanin to the animal. Please justify the reason why they refer to indicaxanthin.

-       Authors should discuss more in depth the molecular components of the extract they believe are responsible for the effects observed in vivo.

-       REF 11 and 12 on the cardiovascular-related, anti-inflammatory effects of betanin and indicaxanthin should be more appropriately substituted by the more recent and relevant ones in 10.1080/10408398.2020.1822277 and 10.1016/j.redox.2014.07.004

-       The vivo study in 10.29219/fnr.v62.1262 should be cited in the introduction where the authors comment the experimental in vivo, in vitro and clinical studies on the fruit of Opuntia spp.

-       REF 15 on the anti-proliferative effects of indicaxanthin should be more appropriately substituted by the more recent and relevant ones in 10.1016/j.phymed.2018.09.171

Author Response

First of all, we would like to thank the reviewers and editors for their thorough evaluation of our manuscript.

Please see the attachment where we provide a point by point response to all comments raised by the reviewer.

Since reviewer 2 asked for English edition, we submitted the article to the English edition service from MDPI and we are waiting for the corrected file.

Reviewer 2 Report

Interesting experimental approach but of little clinical importance- Substantial revision is needed.

Major points:

1. Title is without details, rats and experimental design.  Must be included.

2.  Introduction ref 27 is misleading, implying studies in humans with DILI due to DF. Eliminate this reference and condense the first sentence, focusing on DILI by DF and on published DILI cases assessed for causality using RUCAM,  Cheeck GOOGLE and a report on 81,856 DILI cases published worldwide  and assessed by RUCAM, here you may find some references, and quote all.

3. You mention ferric reduuing activities, but did not discuss this point in dtatail regarding ferroptosis, Does ferroptosis and how play a role in this special DILI? A role for potential treatment by iron chelators? Must be discuussed. Discuss also the origin of the term "ferroptosis". Ferro is clear, but ptosis?

4. Add a apara of limitations of yout study,

Author Response

First of all, we would like to thank the reviewers and editors for their thorough evaluation of our manuscript.

Please see the attachment where we provide a point by point response to all comments raised by the reviewer.

As the reviewer suggested, we submitted the article to the English editing service, at the moment we are waiting for the corrected file.

Round 2

Reviewer 2 Report

Perfect revision.